# Artemether Attenuates Gut Barrier Dysfunction and Intestinal Flora Imbalance in High-Fat and High-Fructose Diet-Fed Mice

**DOI:** 10.3390/nu15234860

**Published:** 2023-11-21

**Authors:** Xinxin Ren, Jia Xu, Ye Xu, Qin Wang, Kunlun Huang, Xiaoyun He

**Affiliations:** 1Key Laboratory of Precision Nutrition and Food Quality, Key Laboratory of Functional Dairy, Ministry of Education, College of Food Science and Nutritional Engineering, China Agricultural University, Beijing 100083, China; 2College of Animal Science and Technology, China Agricultural University, Beijing 100193, China; 3Key Laboratory of Safety Assessment of Genetically Modified Organism (Food Safety), The Ministry of Agriculture and Rural Affairs of the P.R. China, Beijing 100083, China

**Keywords:** artemether, metabolic disorder, gut microbiota, intestinal barrier dysfunction, inflammation

## Abstract

Intestinal inflammation is a key determinant of intestinal and systemic health, and when our intestines are damaged, there is disruption of the intestinal barrier, which in turn induces a systemic inflammatory response. However, the etiology and pathogenesis of inflammatory diseases of the intestine are still not fully understood. Artemether (ART), one of the artemisinin derivatives, has been widely used to treat malaria. Nevertheless, the effect of ART on intestinal inflammation remains unclear. The present study intended to elucidate the potential mechanism of ART in diet-induced intestinal injury. A high-fat and high-fructose (HFHF) diet-induced mouse model of intestinal injury was constructed, and the mice were treated with ART to examine their role in intestinal injury. RT-qPCR, Western blotting, immunohistochemical staining, and 16S rRNA gene sequencing were used to investigate the anti-intestinal inflammation effect and mechanism of ART. The results indicated that ART intervention may significantly ameliorate the intestinal flora imbalance caused by the HFHF diet and alleviate intestinal barrier function disorders and inflammatory responses by raising the expression of tight junction proteins ZO-1 and occludin and decreasing the expression of pro-inflammatory factors TNF-α and IL-1β. Moreover, ART intervention restrained HFHF-induced activation of the TLR4/NF-κB p65 pathway in colon tissue, which may be concerned with the potential protective effect of ART on intestinal inflammation. ART might provide new insights into further explaining the mechanism of action of other metabolic diseases caused by intestinal disorders.

## 1. Introduction

A growing body of research has shown that gut flora is linked to various health issues, particularly metabolic syndrome, diabetes, and obesity [1]. In recent years, a series of studies have reported that a high-fat diet (HFD) can alter the composition of intestinal microbes and their metabolites, promote energy metabolism disorders, and then lead to intestinal barrier damage [2,3]. High fructose consumption appears to be increasing over time, and it has a greater impact on the immune system than glucose consumption [4]. The HFHF diet can change the composition of intestinal flora, reduce the richness of *Akkermansia* and *Bacteroides* in the gut, increase intestinal permeability, and aggravate neutrophil infiltration in the mouse intestine [5,6,7].

The intestine is the largest barrier between organisms and environment, protecting the body from harmful external substances. The intestinal barrier mainly consists of four parts, namely physical, chemical, immunological, and microbiological barriers, among which the physical barrier is an intact intestinal epithelium structure consisting of intestinal mucosal epithelial cells through tight junction proteins, such as occludin, claudins, zonula occludens (ZOs), and tricellulin [8,9]. Food, especially high-fat and high-fructose food, can cause great damage to the intestinal barrier [10]. New shreds of evidence suggest that excess fructose intake may lead to reduced intestinal tight junctions (TJs) and adherens junctions (AJs) expression, expanded intestinal permeability and microbial translocation in the lumen, and elevated lipopolysaccharide (LPS) in the circulatory system [11,12]. LPS is the main component of Gram-negative bacilli’s cytoderm, and it has been shown that intestinal microbial disturbances promote the release of LPS, which in turn damages the intestinal barrier and leads to systemic chronic low-severity inflammation in the host [13,14,15]. Notably, TLR4 is a major receptor molecule in the NF-κB pathway activated by LPS, thereby causing the progression of intestinal inflammation through the release of pro-inflammatory cytokines [16,17]. These latest developments provide new insights into the search for functional components that prevent and treat intestinal inflammation. 

For the past few years, natural products and plant extracts have provided a new therapeutic method, and some clinical trial research has also proved that traditional Chinese medicine has great potential to ameliorate intestinal inflammation [18]. Artemisinin is a sesquiterpene lactone compound with a distinctive oxygen bridge structure distilled from *Artemisia annua* L. [19]. ART is a lipid-soluble derivative of artemisinin, with a stronger antimalarial effect than artemisinin, and has the characteristics of high efficiency and low toxicity. Our team’s previous study found that ART can significantly ameliorate hepatic lipid deposition and reduce inflammatory damage [20]. However, for the moment, there are few studies on the role of ART in intestinal diseases, especially the impact of ART on gut flora and inflammation under the condition of an HFHF diet. Hence, the purpose of this research was mainly to explore the regulatory effect of ART on intestinal microbial disorders, increased intestinal permeability, and intestinal inflammation in HFHF-fed mice.

## 2. Methods

### 2.1. Materials

Artemether was obtained from Macklin, Shanghai, China (Article No. A800587, purity: HPLC ≥ 98%). A high-fat feed (60% of energy from fat) was obtained from Beijing Huafukang Co., Ltd. (Article No. H10060, Beijing, China). Fructose was obtained from Merck, Rahway, NJ, USA (F3510, C_6_H_12_O_6_, HPLC ≥ 99%).

### 2.2. Animal Models

This experiment was approved by the Animal Experiment Ethics Committee of China Agricultural University (approval ID: Aw10401202-4-4, approval time: 1 April 2021). Purchased 6-week-old male C57BL/6J mice from Beijing Vital River Laboratory Animal Technology Co., Ltd. (Beijing, China) were raised in the SPF level animal house of China Agricultural University (12 h light–dark cycle, 22 ± 2 °C). The model of diet-induced intestinal inflammation was established according to previous studies [21,22]. All mice were fed adaptively for 1 week; the twenty-four mice were randomly divided into four groups and fed for 17 weeks (*n* = 6/group): the CHOW group (normal chow diet), HFHF group, ART-L group, and ART-H group (60% HFD +30% (*w*/*v*) fructose water). From the 11th week, the ART-H and ART-L groups were given 50 and 10 mg/kg BW/d doses of artemether by gavage, respectively, for 7 weeks (Figure 1A). ART was dissolved in 0.5% Carboxymethylcellulose sodium (CMC-Na), while mice in the CHOW and HFHF groups were orally administered 0.5% CMC-Na as a placebo.

Fecal samples from mice were collected on the 7th day before the end of the experiment and flash-frozen at −80 °C for subsequent intestinal microbial testing. When the experiment was finished, mice were fasted for 6 h and then immediately sacrificed to collect their blood, colons, spleens, etc. Colonic tissue was flushed with physiological saline, and the contents were squeezed out, then fixed with 4% paraformaldehyde until subsequent experiments. All the samples were stored at −80 °C.

### 2.3. Determination of Serum Parameters

Mouse serum was collected to conduct biochemical analysis, and aspartate aminotransferase (AST), alanine aminotransferase (ALT), high-density lipoprotein (HDL), low-density lipoprotein (LDL), and cholesterol (CHO) were measured (98640000, Indiko Plus fully Automatic Biochemical Analyzer, Thermo Fisher Scientific Inc., Waltham, MA, USA) [23]. LPS of the serum was measured using ELISA kits (CUSABIO, Wuhan, China). OD values were obtained at 450 nm (1410101, Multiskan™ FC Microplate Reader, Thermo Fisher Scientific Inc., Waltham, MA, USA) and analyzed with GraphPad 8 software.

### 2.4. Hematoxylin-Eosin (H&E) Staining

Distal colon tissue immersed in paraformaldehyde fixative was embedded in paraffin, sectioned with a microtome (SLEE, Mainz, Germany), and finally stained with hematoxylin-eosin. The degree of damage to colon tissue was observed under light microscopy (DM2500, Leica, Wetzlar, Germany).

### 2.5. Immunohistochemical (IHC) Staining

Sections of colon tissue (5 µm thick) were deparaffinized in xylene and then hydrated with varying gradients of alcohol and water. Hydrogen peroxide solution and goat serum were used to inactivate the endogenous enzyme activity and block non-specific sites in the sections. F4/80 (ab300421), CD11c (GB11059), ZO-1 (ab276131), and occludin (GB111401) were the detection objects. After washing with phosphate-buffered saline for 30 min, incubation with HRP-conjugated secondary antibodies was performed. The images were observed under a light microscope, and the expression level of target proteins (brown stained area) was analyzed via Image Pro Plus 6.0 software.

### 2.6. Real-Time Quantitative PCR

DNA was extracted from colon tissue using TRIZOL (Introvigen, 15596018, Carlsbad, CA, USA), chloroform, and isopropyl alcohol in sequence. cDNA was synthesized using a specialized kit (TransGen, Beijing, China) [24]. Then, the FastKing One-Step RT-PCR Kit was used to measure the mRNA level of the target gene (TIANGEN Biotech, Beijing, China). *β-actin* was chosen as an internal reference gene. The primer sequences used in this study were obtained from the PrimerBank official website and are shown in Table 1. 

### 2.7. Western Blotting

The colon tissue frozen at −80 °C was ground in liquid nitrogen, then centrifuged at 2000× *g* at 4 °C for 20 min, and the supernatant was taken as the protein to be tested. The protein concentration was determined using the BCA protein assay kit (Beyotime Biotechnology, Beijing, China). Protein samples (20–30 µg) were electrophoresed on 12% SDS-PAGE gel using a Bio-Rad electrophoresis instrument (Bio-Rad, Hercules, CA, USA) and then transferred to a methanol-activated 0.45 um PVDF membrane. The samples were washed for 2 h in a 5% skim milk powder solution prepared with TBST and incubated with an antibody to be tested overnight at 4 °C. β-tubulin (Beyotime Biotechnology, Beijing, China), TLR4, NF-κB p65, MyD88 (Proteintech, Wuhan, China), Phospho-NF-κB p65, TNF-α (Cell Signaling Technology, Danvers, MA, USA), and IL-1β (Abcam, Waltham, MA, USA) were the primary antibodies used in this study. Subsequently, the samples were incubated with a horseradish peroxidase-labeled mouse or rabbit secondary antibody (Abcam, Waltham, MA, USA) for 2 h at room temperature. A chemiluminescence imager was used to take pictures of the target protein band and perform a quantitative analysis of protein expression.

### 2.8. Gut Microbiota Analysis

Total genome DNA from samples was extracted using the CTAB/SDS method. DNA concentration and purity were monitored on 1% agarose gel. A particular primer (16S V4: 515F-806R) was used to amplify the 16S rRNA genes of distinct regions (16S V4/16S V3/16S V3-V4/16S V4-V5) using a barcode [25]. Beta diversity analysis used PAST software (version 2.17) for non-metric multidimensional scaling (NMDS) plotting and was based on the Bray–Curtis operation. The Galaxy online platform was used to perform LEfSe analysis to compare the differential bacterial flora between groups. A correlation cluster analysis of microorganisms with physical and chemical indicators was conducted on the online platform Lianchuan Biocloud.

### 2.9. Statistical Analysis

All data were calculated on GraphPad 8 (GraphPad Software Inc., La Jolla, CA, USA) software using a one-way analysis of variance (Tukey’s multiple comparisons), and the results were displayed as the mean ± SD. When *p* < 0.05, there is a statistical difference. 

## 3. Results

### 3.1. Artemether Suppresses METABOLIC Disorders in HFHF Diet-Fed Mice

Starting from week 11, the treatment group was given different doses of ART (10 and 50 mg/kg) for 7 weeks to analyze the influence of ART on body weight and blood-lipid level of HFHF diet mice (Figure 1A). The results showed that after feeding mice an HFHF diet for 10 weeks, body weight significantly increased. Both doses of ART significantly inhibited body weight increase from week 11 to week 17 (Figure 1B). Further comparing the weight differences of mice in each group in the last week, it was found that ART treatment can effectively reverse the weight gain caused by an HFHF diet (Figure 1C). In addition, the ALT/AST ratio, TC, LDL, and HDL in the serum of mice fed an HFHF diet increased significantly, and ART intervention reversed the above changes (Figure 1D–G). The above results demonstrate that ART can suppress metabolic disorders and weight gain in mice induced by an HFHF diet.

### 3.2. Artemether Regulates the Flora Diversity of HFHF Fed Mice

Accumulating evidence suggests that diet is an important driver of intestinal flora alteration [26,27,28]. Therefore, the feces of mice in each group were collected, and 16S rRNA sequencing was performed to analyze the changes in intestinal flora. Alpha- and beta-diversity indices were used to analyze and compare the effects of daily ART intervention on the diversity of the intestinal microbiota of mice for 7 consecutive weeks. As illustrated in Figure 2A, Shannon’s diversity index was found to be lower in the HFHF group compared to the CHOW group, which was restored after ART intervention. In addition, a significant decrease in the Simpson index was found in the HFHF group, and we observed significant differences only in the ART-L (10 mg/kg) treatment group (Figure 2B); this shows that artemether treatment can significantly improve the microbial composition of mice fed an HFHF diet. 

Beta-diversity analyses, including NMDS, were performed to compare the difference in intestinal flora between groups; compared to the CHOW group, the HFHF group was significantly dispersed on the horizontal MDS1 axis (Figure 2C), indicating that the HFHF diet can induce disturbances of the intestinal microbial structure. Although the HFHF group and the ART group had some overlap on the MSD1 and MSD2 axes, a unique clustering between the ART groups and the HFHF group could be observed. Taken together, alpha and beta diversity indices demonstrated that long-term artemether intake modulates changes in bacterial richness induced by HFHF diets.

### 3.3. Artemether Regulates the Microbiota Composition of the HFHF Fed Mice

To further explore specific transformations in microbial communities, we first analyzed the 10 most abundant bacterial phyla in the four groups at the phylum level through a correlation abundance plot. At the phylum level, *Firmicutes* and *Bacteroidetes* were the two main classifications, accounting for more than 80% of the total sequences. Compared with the CHOW group, the relative richness of *Firmicutes* was significantly increased, and the relative abundance of *Bacteroidetes* was decreased in mice on the HFHF diet. We also found that HFHF-diet increased the *Firmicutes*/*Bacteroidota* ratio. In contrast, ART intervention (10 mg/kg) effectively reversed these transformations (Figure 2D,E). 

The groups mainly presented 16 predominant bacterial genera, including Parabacteroides, Oscillibacter, Lactococcus, Lachnoclostridium, Ileibacterium, GCA-900066575, Faecalibaculum, Erysipelatoclostridium, Dubosiella, Coriobacteriaceae_UCG-002, Colidextribacter, Blautia, Bacteroides, Anaerotruncus, Alloprevotella, and Alistipes. The HFHF group exhibited a higher abundance of Oscillibacter, Alistipes, Colidextribacter, etc., than the CHOW group. Further, the CHOW group presented a high abundance of Alloprevotella relative to the HFHF group (Figure 2F,G). 

Next, the dominant bacterial groups in each treatment group were further explored through LEfSe analysis. As shown in Figure 3, the groups mainly displayed bacteria with significant differences from phylum to species level. *Ruminococcaceae*, *NK4A214_group*, *Peptococcaceae*, *Anaerovorax*, *UBA1819*, and *Rikenellaceae_RC9_gut_group* were dominant bacteria in the ART-L group. However, *Streptococcaceae*, *Bacteroides*, *Lachnospiraceae*, *Colidextribacter*, and *Proteus* were the predominant bacteria in the HFF group. In addition, LEfSe analysis showed that the dominant bacteria of mice in the ART intervention group were significantly different compared with the HFHF group; this may indicate that ART intervention caused significant changes in the dominant bacteria of mice fed the HFHF diet.

### 3.4. Artemether Improves Intestinal Barrier Dysfunction on HFHF-Fed Mice

Intestinal microbial imbalance can promote the release of endotoxin LPS, further impair intestinal barrier function, and induce local inflammation. Thus, the impact of ART on gut barrier dysfunction was assessed by measuring the expression of relevant tight junction proteins. As shown in Figure 4A, compared with the CHOW group, serum LPS levels in the HFHF group were significantly increased, and ART treatment (10 mg/kg) significantly decreased serum LPS levels. Tight junction proteins (TJs) play a key role in intestinal barrier permeability. We detected the levels of occludin and ZO-1 through qRT-PCR (Figure 4B,C). The expressions in the model group were both significantly reduced, but there was no statistical difference in occludin. For the two genes, the downregulation was statistically reversed after ART (10 mg/kg) intervention. In addition, we found that Claudin-2, which accelerates intestinal barrier leakage, was increased under HFHF diet induction compared to the CHOW group and was distinctly reversed after ART intervention (Figure 4D). IHC staining results of ZO-1 and occludin were consistent with the gene level (Figure 4E,F). Furthermore, H and E staining of colon tissue revealed that ART intervention relieved the intestinal epithelial tissue damage and lymphocyte infiltration induced by the HFHF diet (Figure 4G). Therefore, the findings disclosed that ART attenuated the increased intestinal permeability and intestinal barrier damage caused by an HFHF diet.

### 3.5. Artemether Regulates the Expression of Pro-Inflammatory Factors

Inflammation of the gut is known to be critical to the progression of disease. We further evaluated the effect of ART on intestinal inflammation. In Figure 5A–C, the mRNA expression levels of inflammatory signaling factors TNF-α, IL-1β, and IFN-γ in the HFHF group were significantly increased compared to the CHOW group. In contrast, ART treatment effectively reversed this phenomenon. It has been demonstrated that over-activated macrophages are closely associated with the development of intestinal inflammation [29]. Considering that ART can inhibit the expression of LPS and inflammatory factors, we further assessed whether macrophage M1 typing was affected. Results showed that mRNA expressions of Chemokine 4 (CCL4), intercellular cell adhesion molecule-1 (Icam1), and iNOS and CD11c in the HFHF treatment group were significantly increased and were inhibited by ART treatment (Figure 5D–G). Moreover, F4/80 IHC staining showed that the HFHF diet significantly promoted macrophage recruitment, whereas ART intervention significantly reduced macrophage infiltration (Figure 5H). ART also abrogated the HFHF diet-induced increase in CD11c, a marker of macrophage M1 typing (Figure 5I). Surprisingly, macrophage M2 typification did not improve with ART treatment (Appendix A). To sum up, ART ameliorated inflammation levels in colonic tissues while being able to inhibit macrophage recruitment and M1-type polarization, and interestingly, ART treatment was dose-dependent in ameliorating intestinal inflammation in mice. 

### 3.6. Artemether Inhibits LPS/TLR4/NF-κB Signaling Pathway Activation

The previous results demonstrate the improvement of ART on the intestinal flora, intestinal barrier, and LPS levels. We further explored the potential molecular mechanisms of ART-mediated inflammation inhibition in the LPS/TLR-4/NF-κB signaling pathway. Western blot results showed that the HFHF diet induced the activation of the classic NF-κB signaling pathway involving MyD88 in mouse colon tissue, phosphorylated NF-κB p65, NF-κB p65, and increased expression levels, thereby promoting the downstream inflammatory factor IL-1β and TNF-α release. ART treatment reversed this trend, returning the expression levels of these proteins to near-normal levels, and the effect was dose-dependent (Figure 6A–G). The NF-κB signaling pathway in the colon tissue of HFHF-fed mice was activated by LPS released by the imbalance of intestinal flora. We demonstrated that ART can inhibit the activation of the NF-κB pathway, which might be critical to suppressing intestinal inflammation and macrophage recruitment.

### 3.7. Correlation Analysis of Physical and Chemical Indicators with Intestinal Flora

In order to explore the interactive relationship between the gut microbiota and related physical and chemical indicators in mice induced by an HFHF diet, a correlation analysis was conducted by Spearman. *Lactococcus*, *Oscillibacter*, *Bacteroides*, *Anaerotruncus*, and *Colidextribacter* were positively correlated with LDL and CHO; *Alistipes* and *Lachnoclotridium* were positively correlated with final body weight and HDL (Figure 7A). Moreover, *Rikenellaceae_RC9_gut_group* and *Alloprevotella* exhibited a negative correlation with CD11c and CCL4, respectively; in addition, *NK4A214_group* was also negatively correlated with CCL4, IL-1β, and IFN-γ. At the same time, the *Firmicutes*/*Bacteroidota* ratio was positively correlated with the inflammatory factors IL-1β and IFN-γ (Figure 7B). Further analysis revealed that *Oscillibacter*, *Alistipes*, *Colidextribacter*, and *Anaerotruncus* positively correlated with parameters related to intestinal inflammation and were further downregulated by ART treatment (Figure 2G). The level of *Alloprevotella* enrichment was also significantly increased after ART treatment (Figure 2G). These results suggest that changes in intestinal microbial composition may influence the expression of intestinal inflammatory and metabolic parameters in mice.

## 4. Discussion

Excessive intake of HFHF diets is not only a major cause of metabolic diseases but also a risk factor for inducing intestinal damage. Intestinal damage can exacerbate systemic inflammation, so it is critical to investigate drugs that improve the intestinal barrier and inflammation. ART is a derivative of artemisinin. Previous studies have shown that ART inhibits DSS-induced intestinal fibrosis in mice with experimental colitis [30], but there is no reported effect on improving intestinal inflammation and intestinal barrier dysfunction. In particular, it is not clear whether ART in the HFHF diet can reduce intestinal damage and its underlying mechanisms. In this study, we found that ART treatment notably ameliorated metabolic disorders in HFHF diet-fed mice. Furthermore, ART intervention modulated intestinal microbial imbalance, improved intestinal barrier dysfunction, reduced intestinal pro-inflammatory factor levels, and attenuated macrophage M1-type polarization, which may be relevant to the fact that ART regulates gut flora and thereby promotes endotoxin to drive the TLR4/NF-κB p65 pathway. This study is the first to find that ART can alleviate intestinal barrier dysfunction and intestinal inflammation caused by the HFHF diet by regulating intestinal microbial imbalance.

Gut microbes are critical in the development of dietary disorders and have been proven to be involved in leaky gut and obesity-related metabolic diseases [25]. We found that the HFHF diet decreased the richness and diversity of gut microbes, consistent with previous studies [31]. ART treatment increased the diversity of the intestinal microbiome and significantly enhanced microbiota richness. In contrast, *Alistipes* and *Bacteroides* in the HFHF group showed relatively increased proportions, and it was confirmed that *Alistipes* and *Bacteroides* can produce LPS [31,32]. Other than that, the HFHF group showed higher relative richness of *Oscillibacter* and *Colidextribacter* than the CHOW group. Notably, *Oscillibacter* and *Colidextribacter* are reported to be inflammation-associated bacteria that are highly colonized in the intestinal tract of colitis mice [33,34]. By contrast, ART-treated HFHF mice carried a higher abundance of beneficial intestinal bacteria, such as *Alloprevotella*, *Ruminococcaceae*, *NK4A214_group*, and *Dubosiella*. *Alloprevotella* improves host immunity, maintains intestinal barrier integrity, and prevents ulcerative colitis from getting worse [35]; *Ruminococcaceae*, *NK4A214_group*, and *Dubosiella* produce short-chain fatty acids that protect the intestinal mucosa integrity and alleviate the body’s inflammatory response [36,37]. 

Intact epithelial cells are tightly connected to form the physical barrier of the intestine, and tight junction proteins are essential elements of the intestinal barrier. Dysbiosis of gut microbes has been reported to lead to intestinal barrier dysfunction and downregulation of TJ protein expression, which may be related to elevated LPS levels [38]. In the present study, the HFHF diet upregulated serum LPS levels and significantly downregulated the mRNA and protein levels of tight junction proteins ZO-1 and occludin. In addition, we observed that the HFHF diet significantly upregulated mRNA levels of Claudin-2, the first tight junction protein identified to reduce trans-epithelial resistance when overexpressed and have elevated expression when the intestinal barrier is compromised [39,40]; ART treatment restored the levels of these. In this study, we found that the low-dose treatment group was more significant in restoring tight junction protein expression, indicating that ART can successfully protect the intestinal epithelial barrier up to a certain dose, thus inferring that the protective effects of ART on the intestinal mucosal barrier were not entirely dependent on the therapeutic dose.

The intact intestinal barrier can effectively prevent exogenous toxic substances from entering the bloodstream, but once this barrier is broken, LPS, a derivative of Gram-negative bacteria, enters the bloodstream and induces inflammation [41,42]. LPS can interact with immune cell surface receptors such as TLR4, resulting in the activation of the NF-κB pathway via its downstream myeloid differentiation factor MyD88, which induces the release of pro-inflammatory cytokines, ultimately driving the inflammatory response [43]. This study shows that the HFHF diet enhanced the expression levels of LPS and pro-inflammatory factors in serum and colon tissue, indicating that the TLR4/NF-κB signaling pathway is activated. Compared with the HFHF group, ART treatment obviously inhibited the expression levels of MyD88 and TLR4 on the NF-κB pathway and the production of downstream pro-inflammatory factors TNF-α and IL-1β. The NF-κB signaling pathway has been confirmed to be a classic pathway related to immunity and inflammation. It also regulates macrophage polarization [44] and produces pro-inflammatory cytokines that recruit more neutrophils, monocytes, and lymphocytes, thereby exacerbating the host inflammatory response [34,45]. In the present study, in the IHC staining of mouse colon tissue, ART treatment inhibited the activation of M1-type macrophages (F4/80, CD11c), as well as significantly reduced the levels of M1-type macrophage markers (iNOS, CD11c) and chemokines (CCL4); this may suggest that ART can selectively attenuate M1-type polarization of pro-inflammatory macrophages to further reduce intestinal inflammation. Notably, we found that ART has no significant promoting effect on macrophage M2-type markers, suggesting that ART may not ameliorate intestinal inflammation through the secretion of anti-inflammatory mediators by M2 macrophages. However, whether ART inhibits the differentiation of macrophages into M1 type by activating the NF-κB p65 signaling pathway and the specific target genes still needs further verification.

In summary, this study showed that ART ameliorated metabolic disturbance and intestinal damage in HFHF diet-fed mice, which may be related to maintaining intestinal permeability and regulating intestinal flora. However, we found that a low dose of ART is more beneficial in regulating intestinal microbial disorders. We suspect that there may be a feedback effect of a high dose of ART in intestinal microbes, and we will explore this issue in more depth in subsequent experiments. Moreover, this study mainly confirmed the improvement effect of ART in chronic intestinal inflammation, which provides an important reference for our group to further research the function of artemether in acute enteritis models, which is also the focus of our subsequent research. In conclusion, we confirmed that artemether may be a promising pharmaceutical for protection against obesity-associated chronic intestinal damage.

## 5. Conclusions

The present study confirmed that ART significantly reduced HFHF diet-induced metabolic disturbance and colonic histological damage. This action may be mediated via modulating intestinal flora, ameliorating gut barrier dysfunction, and inhibiting NF-κB signaling pathway activation via reducing LPS, subsequently restraining macrophage M1 type and reducing the release of colonic inflammatory factors. This study provides momentous evidence for ART treatment of intestinal flora disorder and intestinal damage induced via the HFHF diet, which enriches the use of ART as a therapeutic medicine for intestinal inflammation. 

## Figures and Tables

**Figure 1 nutrients-15-04860-f001:**
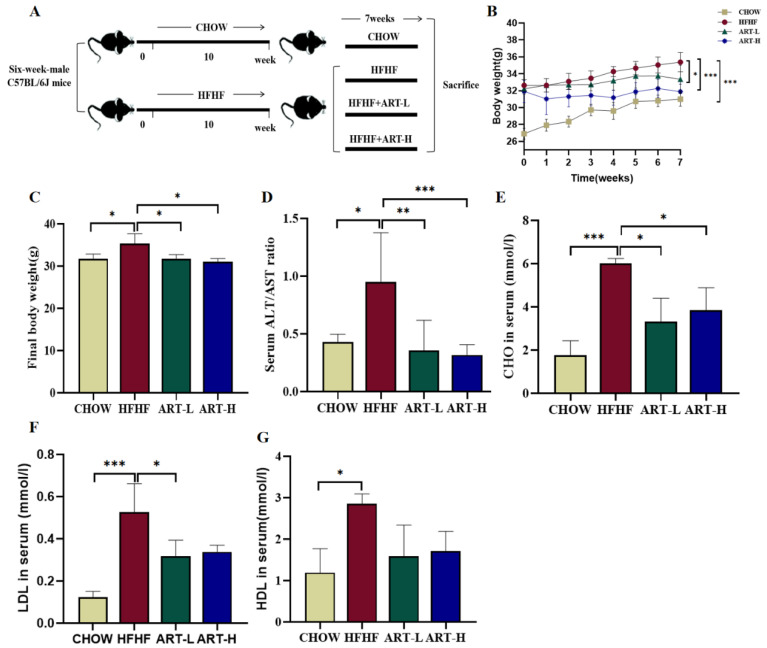
The effect of artemether intervention on metabolic disorders in HFHF-diet mice. (**A**) The flow chart of the animal experiment. (**B**) The effect of ART on weight gain (changes during ART administration). (**C**) Comparison of body weight between groups in the last week. (**D**) ALT/AST ratio in the serum. (**E**–**G**) Lipid content in the serum: TC (**E**), LDL (**F**), and HDL (**G**). Results were shown as mean ± SD (*n* = 6). * *p* < 0.05, ** *p* < 0.01, and *** *p* < 0.001 compared with the HFHF group.

**Figure 2 nutrients-15-04860-f002:**
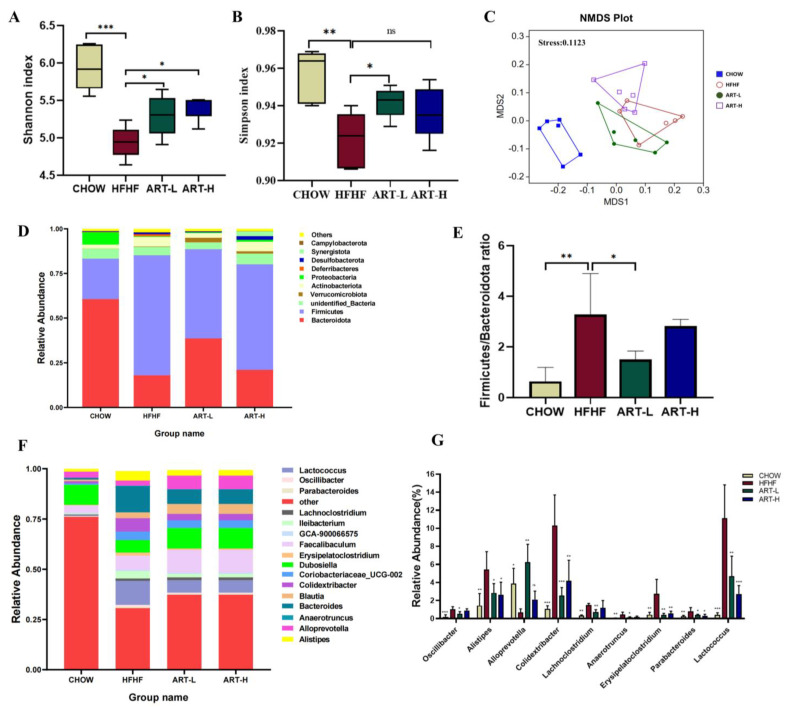
The effect of artemether on the composition of intestinal microbial in the HFHF-diet mice. (**A**,**B**) α-diversity analysis, Shannon’s index (**A**), and Simpson’s index (**B**). (**C**) NMDS plot based on the Bray–Curtis distance. (**D**) Changes in the relative richness of each group of microbiota at the phylum level. (**E**) The ratio of *Firmicutes*/*Bacteroidota*. (**F**) Changes in the relative richness of microorganisms at the genus level. (**G**) Relative richness of 10 significantly altered bacterial genera. Results were shown as mean ± SD (*n* = 6 per group). Results were shown as mean ± SD (*n* = 6). ^ns^ *p* > 0.05, * *p* < 0.05, ** *p* < 0.01, and *** *p* < 0.001 compared with the HFHF group.

**Figure 3 nutrients-15-04860-f003:**
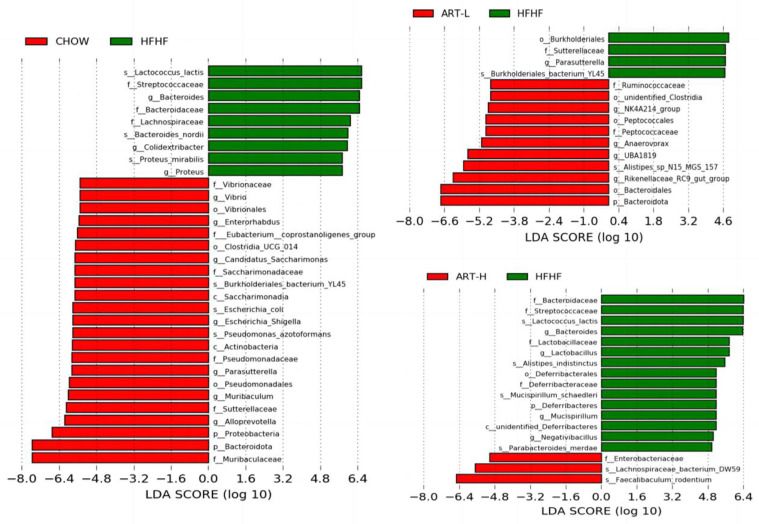
LEfSe analysis. Taxonomy histogram showing four major biomarker taxa. Differentially expressed taxa with an LDA score >4.0. Comparative analysis with the HFHF group, respectively. *p* < 0.05.

**Figure 4 nutrients-15-04860-f004:**
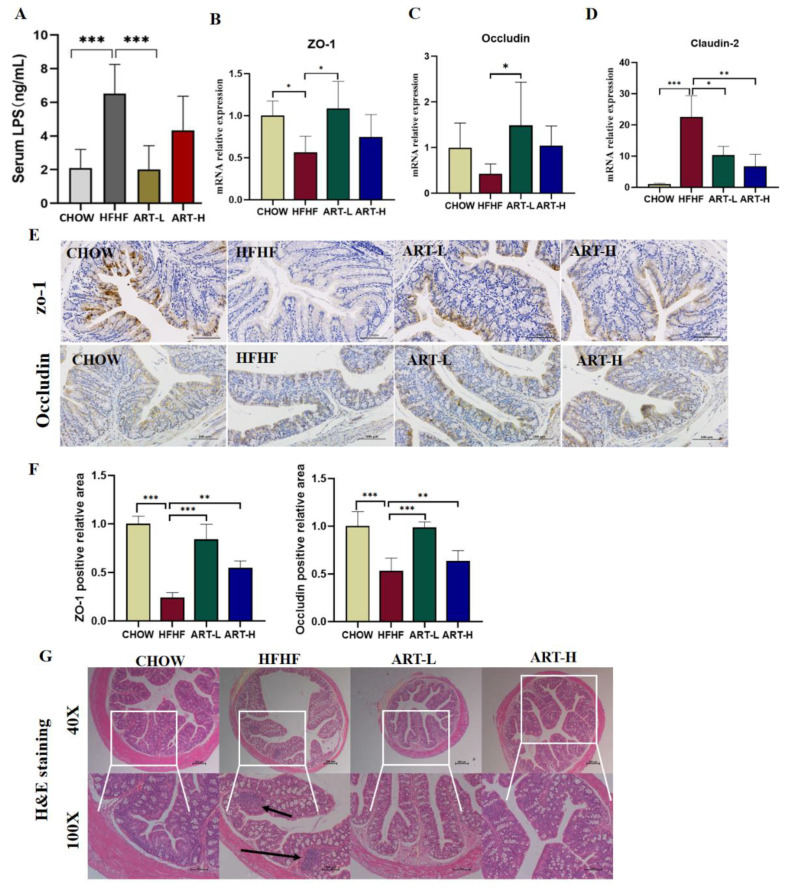
The effect of artemether on the intestinal barrier in the HFHF-diet mice. (**A**) Serum LPS concentration. mRNA levels of ZO-1 (**B**), occludin (**C**), and Claudin-2 (**D**). Immunohistochemistry and quantitative analysis of ZO-1 (**E**) and occludin (**F**) in the colon tissue. (**G**) Histology analysis of the colon based on H and E; image magnification is 40× and 100×; the black arrows show the infiltration of inflammatory cells. Results were shown as mean ± SD (*n* = 6). * *p* < 0.05, ** *p* < 0.01, and *** *p* < 0.001 compared with the HFHF group.

**Figure 5 nutrients-15-04860-f005:**
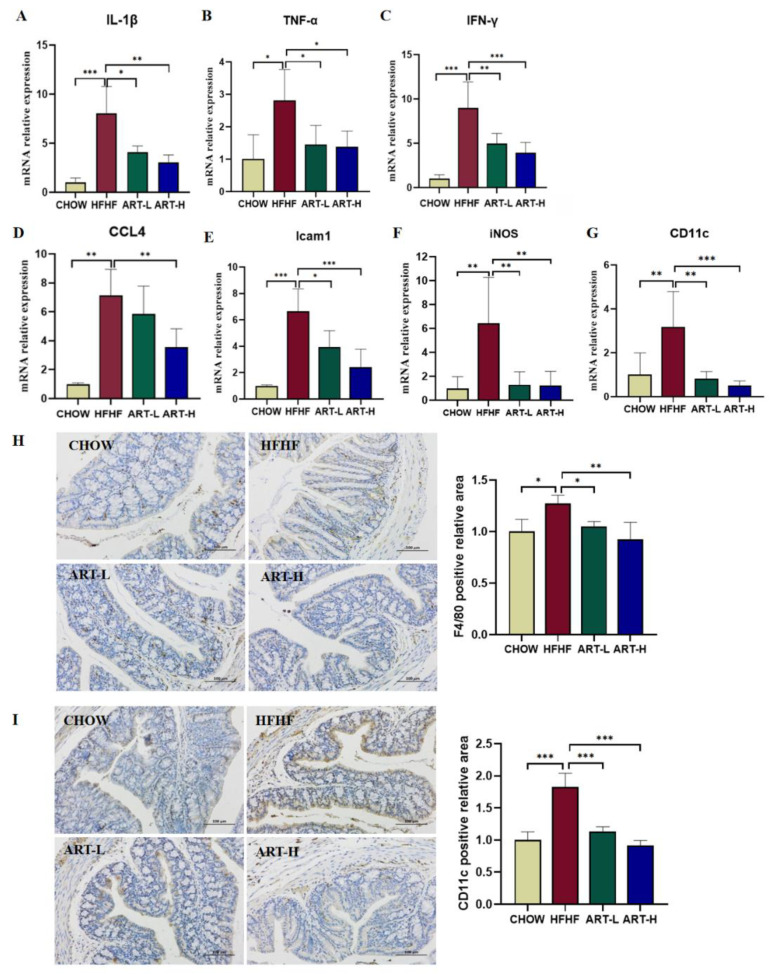
The effects of artemether on pro-inflammatory cytokine production and macrophage polarization. Real-time PCR analysis of inflammatory factor-related genes, (**A**) IL-1β, (**B**) TNF-α, (**C**) IFN-γ, (**D**) CCL4, (**E**) Icam1, (**F**) iNOS, and (**G**) CD11c. Representative immunohistochemical staining images and quantification of F4/80 (**H**) and CD11c (**I**) in mouse colon tissue. Results were shown as mean ± SD (*n* = 6). * *p* < 0.05, ** *p* < 0.01, and *** *p* < 0.001 compared with the HFHF group.

**Figure 6 nutrients-15-04860-f006:**
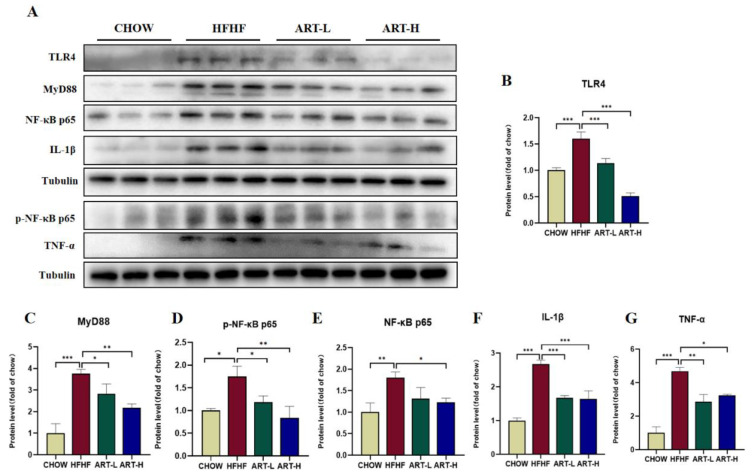
The effect of artemether on the LPS/TLR4/NF-κB-related signaling pathway. (**A**–**G**). Images and quantification of Western blot analysis of TLR4, MyD88, NF-κB p65, phospho-NF-κB p65, IL-1β, and TNF-α. Results were shown as mean ± SD (*n* = 6). * *p* < 0.05, ** *p* < 0.01, and *** *p* < 0.001 compared with the HFHF group.

**Figure 7 nutrients-15-04860-f007:**
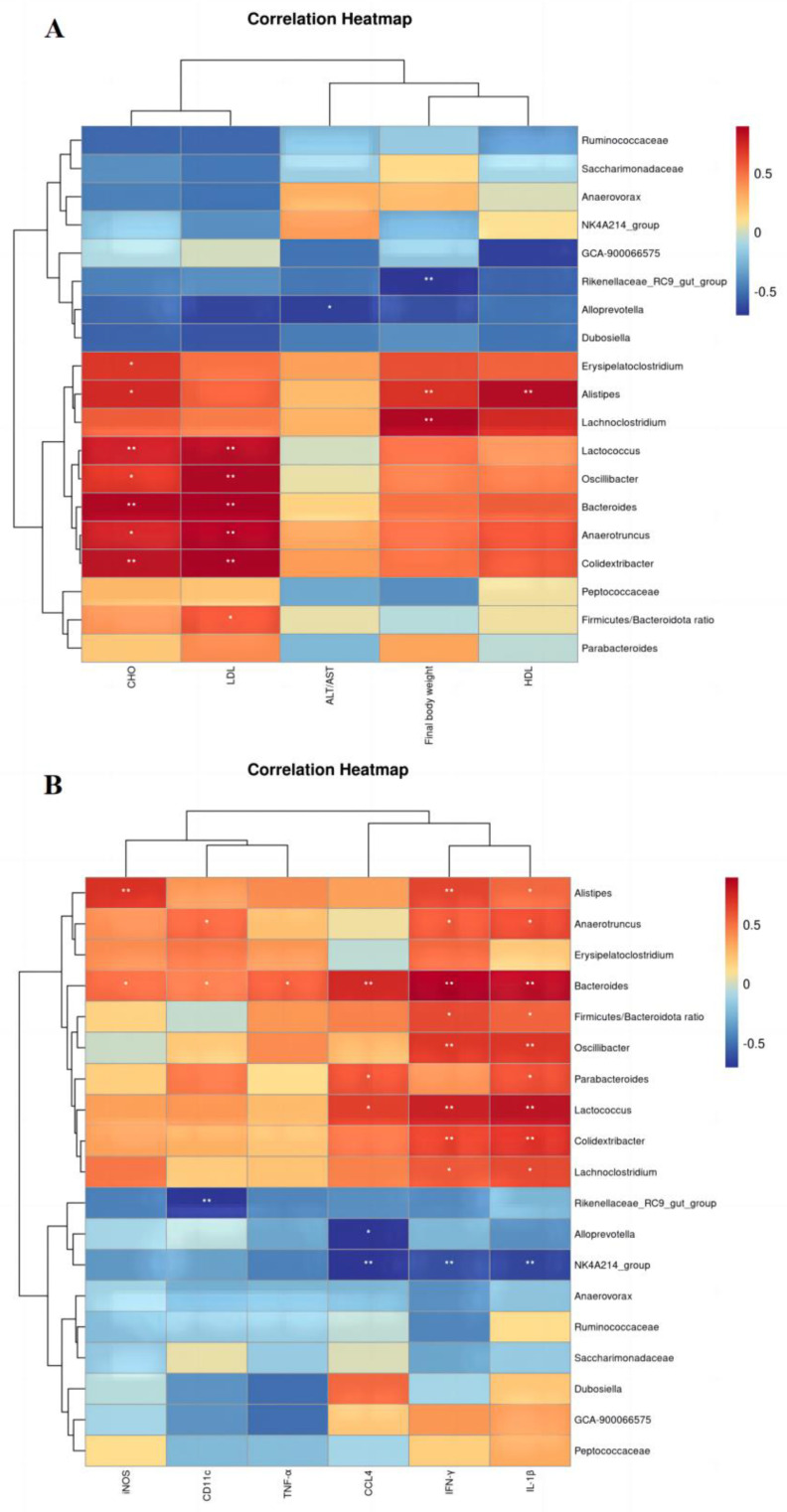
Relevance analysis. Spearman analysis of intestinal flora and parameters. * *p* < 0.05, ** *p* < 0.01 compared with the HFHF group. (**A**) Correlation of key gut microbiota and metabolic parameters. (**B**) Correlation between key gut flora and inflammatory factors.

**Table 1 nutrients-15-04860-t001:** Primer sequences used for RT-qPCR.

Gene	Forward Primer (5′→3′)	Reverse Primer (5′→3′)
*β-actin*	GGGCCGGACTCGTCATAC	CCTGGCACCCAG CAC AAT
*ZO-1*	ACCACCAACCCGAGAAGAC	CAGGAGTCATGGACGCACA
*Occludin*	TTGAAAGTCCACCTCCTTACAGA	CCGGATAAAAAGAGTACGCTGG
*Claudin-2*	CAACTGGTGGGCTACATCCTA	CCCTTGGAAAAGCCAACCG
*iNOS*	GTTCTCAGCCCAACAATACAAGA	GTGGACGGGTCGATGTCAC
*CD11c*	CTGGATAGCCTTTCTTCTGCTG	GCACACTGTGTCCGAACTCA
*IFN-γ*	ATGAACGCTACACACTGCATC	CCATCCTTTTGCCAGTTCCTC
*CCL4*	TTCCTGCTGTTTCTCTTACACCT	CTGTCTGCCTCTTTTGGTCAG
*Icam1*	GTGATGCTCAGGTATCCATCCA	CACAGTTCTCAAAGCACAGCG
*IL-1β*	GCAACTGTTCCTGAACTCAACT	ATCTTTTGGGGTCCGTCAACT
*TNF-α*	GACGTGGAACTGGCAGAAGAG	TTGGTGGTTTGTGAGTGTGAG

## Data Availability

The data presented in this study are available on request from the corresponding author. The data are not publicly available due to laboratory confidentiality policy.

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
