# Peer review of "Artemether Attenuates Gut Barrier Dysfunction and Intestinal Flora Imbalance in High-Fat and High-Fructose Diet-Fed Mice"

_nutrients, 2023, doi:10.3390/nu15234860_

Round 1

Reviewer 1 Report

Comments and Suggestions for Authors

The many studies shown that gut microbiota is associated with many and various health problems, especially metabolic syndrome, diabetes, obesity but also distant from the metabolism and digestive system: neurological health problems according to the recognized brain-intestine-microbiota axis: constant bidirectional communication between the gastrointestinal tract and the brain. A panorama which, as can be seen from these premises, is still to be explored, but which finds explanations on the etiology that is investigated in this research work. This study aim to elucidate the underlying mechanism of Artemether in diet induced intestinal injury. A high fat and high fructose (HFHF) diet-induced mouse model of intestinal injury was constructed and the mice treated with Artemether to examine its role in intestinal injury. 16S rRNA gene sequencing, western blotting, real-time polymerase chain reaction and immunohistochemical staining were used to investigate the effects and mechanisms of Artemether against intestinal inflammation. Intestinal microbial imbalance can promote the release of endotoxin LPS, further impair intestinal barrier function and induce local inflammation. Thus, the effect of Artemether on  gut barrier dysfunction was further evaluated and compared with CHOW group, serum LPS levels in HFHF group were significantly increased, and Artemether treatment significantly decreased serum LPS levels. In particolary they determined the expression levels of Occludin and ZO-1 by qRT-PCR, the mRNA expressions of Occludin and ZO-1 were significantly reduced in the HFHF group. To strengthen the hypotheses stated above of the effect of Artemether on intestinal flora, intestinal barrier and LPS levels, they explored the potential molecular mechanisms of Artemether-mediated inflammation inhibition from the LPS/TLR-4/NF-κB signaling pathway.

This study, in its analysis has not yet been investigated in any line of research visible in the literature, especially at a molecular level in fact, elucidate the underlying mechanism of Artemether in diet induced intestinal injury. A high fat and high fructose (HFHF) diet-induced mouse model of intestinal injury was constructed and the mice treated with Artemether to examine its role in intestinal injury. 16S rRNA gene sequencing, western blotting, real-time polymerase chain reaction and immunohistochemical staining were used to investigate the effects and mechanisms of Artemether against intestinal inflammation

As already stated in the previous point, there are no evident and molecular and immunohistochemical-based studies in the literature that clarify the mechanism of etiopathogenesis

This study provides masterfully momentous evidence for future treatment of intestinal flora disorder and intestinal damage induced via HFHF diet, enriches the use of Artemether as a therapeutic medicine for intestinal inflammation. Therefore a singular work in its workmanship which certainly increases the visibility of the Journal that will host it.

Author Response

Dear Reviewer,

On behalf of my co-authors, we thank you very much for your positive and constructive comments and suggestions on our manuscript entitled “Artemether attenuates gut barrier dysfunction, and intestinal flora imbalance in High-Fat and High-Fructose diet-fed mice” (nutrients-2691608).

We are very grateful to you for your recognition and encouragement of this research.

In this study, we first found that artemether can regulate intestinal microbial disturbances induced by high-fat and high-fructose (HFHF) diets and reduce the production of endotoxin LPS. The production of LPS can lead to the destruction of the intestinal barrier and induce local inflammation. Here, we confirmed that artemether can reverse the intestinal barrier dysfunction induced by HFHF diet through analysis of molecular (mRNA) and protein (immunohistochemistry) levels of tight junction proteins. Furthermore, we analyzed the mRNA expression levels of inflammatory factors and found that artemether can reduce intestinal inflammation induced by HFHF diet. Based on the above findings, we tried to further explain from the mechanism level. Therefore, the more accurate and quantitative Western Blot experiment well explained that LPS can activate the NF-κB signaling pathway, thereby causing an inflammatory response, while artemether can significantly inhibit the NF-κB pathway to exert anti-inflammatory effects.

The molecular level of this study is mainly reflected in the expression at the RNA level.

Thank you very much again for your attention and time. Look forward to hearing from you! 

Reviewer 2 Report

Comments and Suggestions for Authors

Overall, this is an interesting paper that describes the effect of artemether treatment on gut barrier and intestinal flora in HFHF diet-fed mice. The methods are generally appropriate, but there are some errors in the statistical analysis that lessen the potential impact of the findings.

Specifically, the authors use one-way ANOVA to analyze data that could be influenced by multiple variables. For example, the data in Figure 1B (body weight) could be influenced by diet (CHOW vs. HFHF), artemether treatment (with/without), dose of artemether (low vs. high), and time (0-7 days). Therefore, the data should be analyzed with four-way ANOVA. Even if the authors decide to evaluate body weight only on the last day of treatment, the data should still be analyzed with three-way ANOVA. This same problem applies to the data in Figures 1C-G, 2A-B, 4A-D, 4F, 5A-G, and 6B-G. Additionally, the authors should indicate whether the data follow a Gaussian distribution before using parametric statistical tests such as ANOVA.

Another concern is that the authors state in line 41 that HFHF improves gut permeability, while references 5 and 6 indicate that HFHF increases gut permeability and worsens colitis.

Finally, the authors should add the criteria for excluding female mice from the study.

Minor concerns:

  • Line 262: Lack of capital letter.
  • Line 293: Typo.

Author Response

Dear Reviewer,

We thank you very much for their positive and constructive comments and suggestions on our manuscript entitled “Artemether attenuates gut barrier dysfunction, and intestinal flora imbalance in High-Fat and High-Fructose diet-fed mice” (ID: 2691608). We have studied your comments carefully and have made revisions which are marked in yellow in the manuscript. Attached please find the revised version. The specific modifications and responses are as follows:

  1. About the statistical methods of this study

The author’s answer:

Based on your suggestion, we further examined and thought about the statistical analysis methods of this study. The study used one-way ANOVA, the main reason is that only comparisons between groups were conducted, which meets the requirements of one variable in one-way ANOVA. CHOW group and artemether treated groups were compared with the model group HFHF for significant differences. The statistical analysis methods of this study mainly refer to several research articles [1-3].

  1. “Another concern is that the authors state in line 41 that HFHF improves gut permeability, while references 5 and 6 indicate that HFHF increases gut permeability and worsens colitis”

The author’s answer:

We are very sorry for our careless mistake and it was rectified at Line 41, we have corrected the “improve” into “increase”, and marked it in yellow.

  1. “Finally, the authors should add the criteria for excluding female mice from the study”

The author’s answer:

Early research results indicate that male mice of the C57BL/6 strain fed a high-fat-diet (HFD) diet are more susceptible to dietary obesity and related metabolic diseases than females, under a high-fat diet, male mice show greater sensitivity to body weight [4, 5]. Considering that the metabolic diseases of C57BL/6 mice are also induced by diet, we chose male mice for experiments.

  1. “Minor concerns”

The author’s answer:

We sincerely thank for your careful reading. According to your suggestion, we have corrected the “in” into “In”, in line 262, “and” into “And”, in line 293, and marked them in yellow.

Thank you very much again for your attention and time. Look forward to hearing from you!

References

  1. Noval Rivas M, Wakita D, Franklin MK, Carvalho TT, Abolhesn A, Gomez AC, Fishbein MC, Chen S, Lehman TJ, Sato K et al: Intestinal Permeability and IgA Provoke Immune Vasculitis Linked to Cardiovascular Inflammation. Immunity 2019, 51(3):508-521.e506.
  2. Yang JY, Chen SY, Wu YH, Liao YL, Yen GC: Ameliorative effect of buckwheat polysaccharides on colitis via regulation of the gut microbiota. Int J Biol Macromol 2023, 227:872-883.
  3. Zhuang H, Yao X, Li H, Li Q, Yang C, Wang C, Xu D, Xiao Y, Gao Y, Gao J et al: Long-term high-fat diet consumption by mice throughout adulthood induces neurobehavioral alterations and hippocampal neuronal remodeling accompanied by augmented microglial lipid accumulation. Brain Behav Immun 2022, 100:155-171.
  4. Heo M, Pietrobelli A, Wang D, Heymsfield SB, Faith MS: Obesity and functional impairment: influence of comorbidity, joint pain, and mental health. Obesity (Silver Spring) 2010, 18(10):2030-2038.
  5. Peng C, Xu X, Li Y, Li X, Yang X, Chen H, Zhu Y, Lu N, He C: Sex-specific association between the gut microbiome and high-fat diet-induced metabolic disorders in mice. Biol Sex Differ 2020, 11(1):5.
